# Similar Pro- and Antiangiogenic Profiles Close to Delivery in Different Clinical Presentations of Two Pregnancy Syndromes: Preeclampsia and Fetal Growth Restriction

**DOI:** 10.3390/ijms24020972

**Published:** 2023-01-04

**Authors:** Weronika Dymara-Konopka, Marzena Laskowska, Ewelina Grywalska, Anna Hymos, Anna Błażewicz, Bożena Leszczyńska-Gorzelak

**Affiliations:** 1Department of Obstetrics and Perinatology, Medical University of Lublin, 8 Jaczewskiego Street, 20-095 Lublin, Poland; 2Department of Experimental Immunology, Medical University of Lublin, 4a Chodźki Street, 20-093 Lublin, Poland; 3Department of Pathobiochemistry and Interdisciplinary Applications of Ion Chromatography, Biomedical Sciences, Medical University of Lublin, 1 Chodźki Street, 20-093 Lublin, Poland

**Keywords:** ischemic placental syndrome, preeclampsia, fetal growth restriction, soluble endoglin (sEng), soluble fms-like tyrosine kinase-1 (sFlt-1), placental growth factor (PlGF), sFlt-1/PlGF and sFlt-1*sEng/PlGF ratios, pregnancy adverse outcomes

## Abstract

The purpose of this study was to evaluate serum levels of anti- and pro-angiogenic substances measured using enzyme-linked immunosorbent assays and their ratios in pregnancies complicated by different clinical subsets of placental ischemic syndrome: preeclampsia and/or fetal growth restriction. A prospective case-control study was performed consisting of 77 singleton pregnancies complicated by preeclampsia, preeclampsia with concurrent fetal growth restriction (FGR), and isolated normotensive FGR pairwise matched by gestational age with healthy pregnancies. The entire study cohort was analyzed with respect to adverse pregnancy outcomes that occurred. In all investigated subgroups, placental growth factor (PlGF) was lower and soluble endoglin (sEng), the soluble fms-like tyrosine kinase-1—sFlt-1/PlGF and sFlt-1*sEng/PlGF ratios were higher than in the control group. The differences were most strongly pronounced in the PE with concurrent FGR group and in the sFlt-1/PlGF ratio. The highest sFlt-1 values in preeclamptic patients suggest that this substance may be responsible for reaching the threshold needed for PE to develop as a maternal manifestation of ischemic placental disease. The FGR is characterized by an elevated maternal sFlt-1/PlGF ratio, which boosts at the moment of indicated delivery due to fetal risk. We concluded that angiogenic imbalance is reflective of placental disease regardless of its clinical manifestation in the mother, and may be used as support for the diagnosis and prognosis of FGR.

## 1. Introduction

Preeclampsia and fetal growth restriction are considered major pregnancy syndromes that significantly contribute to perinatal morbidity and mortality [1,2,3,4,5]. In many cases, FGR occurs together with PE. They share the challenge that currently there is no effective treatment to stop the progress of clinical deterioration, and it is still delivery that definitely eliminates the direct threat for the mother and fetus during pregnancy, resulting in prematurity-associated consequences for the newborn.

Preeclampsia affects 3–6% of pregnant women worldwide, whereas fetal growth restriction (FGR) is a condition affecting approximately 8% of all pregnancies and contributing to 30% of stillbirths [1,2]. There is a significant overlap in the incidence of PE and FGR, but the association is particularly strong in early-onset disease, diagnosed before 34 weeks, with evidence suggesting that up to 75% of preeclamptic pregnancies will be further complicated by the presence of fetal growth restriction. In contrast, FGR presents simultaneously with late-onset PE in only 10% of preeclamptic cases [3,4,5]. However, concurrent FGR in the course of PE does not lead to a more severe maternal disease phenotype [5,6]. The maternal symptoms of PE do not show any differences between women, regardless of the growth percentiles of the fetus.

Mothers of babies with impaired growth are at increased risk of developing PE [7,8]. However, it has been shown that only approximately 15% of women with FGR will be diagnosed with superimposed preeclampsia later in pregnancy [9]. The management of FGR with concurrent PE should be combined with preeclamptic women’s surveillance, which also takes into consideration the maternal condition and the possibility of rapid fetal deterioration in this disease [10]. 

In PE, an antiangiogenic state has been implicated as a mechanism of disease. Elevated levels of sFlt-1 and sEng and a lower level of PlGF are features of PE and have already established a significant role in the pathogenesis of the condition. They are also associated with maternal endothelium dysfunction and impaired nitric oxide production. The current knowledge in the area of angiogenic balance and dynamics implicates its possible use in clinical practice. Additionally, recent studies have reported similarities in the mechanisms that underlie the development of PE and FGR. They are both defined by some level of placental insufficiency and thus are considered placenta-originating disorders that share many pathological features [11]. Further, similar changes in maternal levels of angiogenic biomarkers are supposed to be found among them [11,12,13,14]. Disordered angiogenesis is one of the processes that is inherently connected with the development of placental insufficiency. The alterations in sFlt-1, PlGF, and sEng levels taken together are believed to describe placental insufficiency better than each marker separately. 

In addition, it is now the subject of research to determine whether the significance of angiogenic factors in PE may be extrapolated to FGR as a part of the clinical picture of placental ischemic disease [15,16]. Although their role has been widely studied in the context of the prediction [17,18,19,20,21,22,23,24,25,26,27,28,29,30] and diagnosis [31,32,33,34,35,36,37] of PE, there is growing evidence that the angiogenic profile is reflective of placental disease and can also be observed in FGR [17,18,38,39,40,41,42,43,44,45,46], especially in patients with abnormal Doppler studies in uterine and umbilical arteries [43]. In PE, maternal and fetal conditions should be kept under surveillance. Tight control of blood pressure in PE results mostly in benefits for the mother [47]. At the same time, it does not improve the rate of adverse perinatal outcomes but may lead to a negative effect on fetal growth without increasing illness or death in the infant. As of yet, in FGR, there is no therapy that improves fetal growth in utero, and current management is to deliver the fetus before intrauterine death or irreversible organ damage occurs [48]. This is particularly challenging in early-onset FGR, where delivery entails additional risks to the baby from extremely preterm birth, with its own attendant short- and long-term complications [49,50,51]. In recent years, better monitoring of PE and FGR has been a subject of growing interest, and the range of diagnostic tools at our disposal involving angiogenic markers is expanding. The evaluation of adverse outcome risk in FGR and PE patients is of particular interest.

Further, a comparison between FGR alone and FGR associated with PE as well as isolated PE regarding levels of maternal sFlt-1, sEng, and PlGF and their ratios could provide valuable information for understanding these different manifestations of placental syndrome. In the definitions of small for gestational age (SGA) as fetal growth below the 10th percentile, which may comprise both small but normally grown babies and intrauterine/fetal growth restriction (IUGR, FGR) with placenta-mediated restricted growth, the terms are often confused, and limited studies in this area have led to inconsistent results. The ongoing advances in terms of the definitions of these two entities—FGR and PE—enable us to precisely select a study group of patients who meet the newest, very strict criteria for both, which were recently formed by international experts and societies to explain differences and similarities in these great obstetrical syndromes [52].

In this study, we hypothesized that FGR and PE demonstrate the same profile in the most widely analyzed pro- and antiangiogenic substances: PlGF, sEng, and sFlt-1, with possible differences in the degree of severity of imbalances. In putting our work into a clinical, practical perspective, the most widely analyzed values of the sFlt-1 were compared to the PlGF ratio in investigated subgroups according to gestational age with preliminary cutoffs and current evidence of the clinical application for the diagnosis, prognosis, and assessment of PE. Finally, we also investigated angiogenic imbalance in the context of adverse outcomes that occured across the entire study population.

## 2. Results

### 2.1. Patient Demographic Data and Clinical Characteristics

There were no statistically significant differences with regard to gravidity and parity, maternal age, weight, or height in patient profiles between groups. In the PE with FGR group, the mean gestational age at delivery was significantly lower than in the PE alone group (median value, 32 vs. 35 weeks), but there were no significant differences in gestational age between other subgroups or any of the subgroups and the control group. The systolic blood pressure (SBP) and mean arterial blood pressure (MAP) values were significantly higher in all investigated subgroups in comparison with control groups, and diastolic blood pressure (BDP) was higher than controls in the two PE groups (PE + FGR and iPE). Aspartate and alanine transferases (AST and ALT), uric acid (UA), and urea values were higher in preeclamptic patients in both groups (PE + FGR and iPE) than in healthy controls.

The patient demographic data and clinical characteristics of pregnant women in all investigated groups are shown in Table 1 and Table 2. Sonographic parameters with Doppler flow evaluation are presented in Table 3.

In our study, 83% (30/36) of patients with FGR had abnormal Doppler study results (UtA or UA above 95 pc or MCA or CPR under 5 pc), and the remaining 17% (6/36) had exclusively EFW under 3 pc. The doppler study analysis revealed a statistically higher uterine artery mean pulsatility index in both FGR groups and a higher umbilical artery pulsatility index in all investigated subgroups compared to controls. It was observed that a positive correlation existed between the sFlt-1/PlGF ratio values and uterine artery PI (R = 0.8, *p* < 0.00005), umbilical artery PI (R = 0.52, *p* < 0.00005), and RI (R = 0.47, *p* < 0.00005) in the entire study population using Spearman’s correlation analysis. These correlations were also observed separately for iFGR and both PE groups together (including iPE with PE + FGR).

Furthermore, with respect to perinatal results, neonatal birth weight was lower than controls in the PE + FGR, iPE, and iFGR groups, which is mostly due to lower gestational age at delivery as all the control patients delivered at term, but birth weight percentiles were significantly lower only in the FGR groups: iFGR and PE + FGR. The patients in the PE + FGR and iPE groups had significantly lower APGAR scores than controls. Additionally, the detailed perinatal outcomes are presented in Table 4. 

### 2.2. Maternal Serum PlGF

The entire study group had significantly lower PlGF values (*p* < 0.001) than the control group. The PlGF serum levels were significantly lower in the iFGR (median 154 pg/mL, quartile range 117–221, *p* < 0.002), iPE (median 142 pg/mL, quartile range 27–227, *p* < 0.001) and PE + FGR groups (median 42 pg/mL, quartile range 22–113, *p* < 0.001) than in the normal pregnancy group (median 769 pg/mL, quartile range 444–1248). The differences between subgroups were not significantly different. 

The maternal serum PlGF levels were significantly higher in healthy pregnant control patients at <34 weeks of gestational age (median 1247 pg/mL, quartile range 1078–1480) than in healthy pregnant women at more advanced gestation (median 500 pg/mL, quartile range 338–769, *p* < 0.005).

### 2.3. Maternal Serum sFlt-1

The entire study group had significantly higher sFlt-1 values (*p* < 0.001) than the control group. The sFlt-1 serum levels were significantly higher only in the preeclamptic subgroups: iPE (median 76,345 pg/mL, quartile range 8614–133,888 pg/mL, *p* < 0.05) and PE + FGR group (median 115,702 pg/mL, quartile range 14,981–221,278 pg/mL, *p* < 0.001) when compared to normal pregnancies (median 8878 pg/mL, quartile range 5574–10,809 pg/mL). The iFGR group presented clearly higher sFlt-1 concentrations than the control group, but the differences were not statistically significant (median 33,590 pg/mL, quartile range 13,871–66,994 pg/mL). The differences between all subgroups were not statistically significant.

The maternal serum sFlt-1 levels were significantly lower in healthy pregnant control patients at <34 weeks of gestational age (median 6567 pg/mL, quartile range 2647–9105 pg/mL) than in healthy pregnant women at more advanced gestation (median 10,170 pg/mL, quartile range 9501–11,407 pg/mL, *p* < 0.05).

### 2.4. Maternal Serum sEng

The entire study group had significantly higher sEng values (*p* < 0.001) than the control group. The sEng serum levels were significantly higher in the iFGR (median 11.7 ng/mL, quartile range 9.1–11.9 ng/mL, *p* < 0.05), iPE (median 11.5 ng/mL, quartile range 10.2–11.9 ng/mL, *p* < 0.003) and PE + FGR groups (median 12.0 ng/mL, quartile range 11.8–12.2 ng/mL, *p* < 0.001) when compared to normal pregnancy group (median 5.8 ng/mL, quartile range 4–8.3 ng/mL). The differences between the studied subgroups of patients with pregnancy complicated by PE and/or FGR were not significantly different.

There was a strong positive correlation between sEng and sFlt-1 levels in the entire study population (R = 0.69, *p* < 0.001).

The maternal serum sEng levels were significantly lower in healthy pregnant control patients at <34 weeks of gestational age (median 4.1 ng/mL, quartile range 3.4–6.0 ng/mL) than in healthy pregnant women at more advanced gestation (median 7.4 ng/mL, quartile range 6.0–9.9 ng/mL, *p* < 0.05).

### 2.5. sFlt-1/PlGF and sFlt-1*sEng/PlGF Ratios

The entire study group had significantly higher ratios (*p* < 0.001) than the control group. All of the investigated subgroups had significantly higher values of both ratios than the control group (*p* < 0.005). The sFlt-1/PlGF ratio was the highest and most pronounced in the iPE and PE with concurrent FGR groups (median values 30 and 100 times higher than controls, respectively). In all three investigated subsets the ratio was significantly higher than that in control subjects. None of the subgroups differed significantly from any other subgroup. At delivery, elevation of the sFlt-1 to PLGF ratio in iFGR, iPE and PE + FGR patients reached extreme, very high values exceeding the preliminary cutoff of 655 in most women (64%, 65% and 70% respectively), with similar occurrence in all investigated subgroups. The distributions of the values of sFlt-1, sEng, PlGF and the sFlt-1/PlGF and sFlt-1*sEng/PlGF ratios in women with isolated PE, isolated FGR, combined PE and FGR and pairwise gestational age-matched healthy controls are represented in Table 5 and Figure 1.

Forty-eight hours before delivery, elevation of the sFlt-1 to PlGF ratio in iFGR, iPE and PE + FGR patients reached extreme values exceeding the cutoff of 655/201 in most women (64%, 65% and 70% respectively) with similar occurrence in all investigated subgroups. The results are presented in Table 6.

### 2.6. Adverse Pregnancy Outcomes

Finally, the angiogenic disturbances were checked in the group with adverse (N = 23) and no adverse outcomes (N = 34) were observed among the study cohort. There were 41 adverse events in total, which occurred in 23 patients from the entire study group (40%, 23/57). Most of the events occurred in PE + FGR subgroup (59%, 24/41) which is also the group with the statistically lowest gestational age. A detailed list of adverse outcomes and their occurrence in each study subgroup as well as their relation to the proposed cutoff value of 655 for adverse outcomes increased risk are listed in Table 7. At the moment of prompt delivery, there were no significant differences in terms of angiogenic factor levels or their ratios between the adverse and no-adverse groups.

**Table 7 ijms-24-00972-t007:** Adverse outcomes, their occurrence in the PE + FGR, iPE and iFGR groups and relation to the sFlt/PlGF ratio cutoff values.

	Adverse Outcome (AO)	Total	PE + FGR	iPE	iFGR
	PATIENTS WITH AO	23/57	10/22	11/21	2/14
	Ratio > 655 in the AO group	10/23	6/10	3/11	½
	% with the ratio > 655 in AO group	44%	60%	27%	50%
	sFlT-1/PlGF median value in AO group	476	840	446	537
	sFlT-1 /PlGF mean value in AO group	2610	3776	1926	537
	ANY ADVERSE EVENT	41 (100%)	24 (59%)	15 (37%)	2 (4%)
	**Maternal**				
1	HELLP syndrome	8	5	3	0
2	Placental abruption	3	2	0	1
3	Blood product transfusion	5	3	2	0
4	Neurological symptoms (eclamptic fits, visual disturbances, intense headache)	9	3	6	0
5	Renal oliguria or AKI	3	1	2	0
6	Intravenous antihypertensive therapy with 3rd drug	9	7	2	0
	**Fetal**				
5	Not viable fetus/ IUD	4	3	0	1

HELLP—hemolysis, elevated liver enzymes, low platelet syndrome; AKI—acute kidney injury; IUD—intrauterine demise. The results of sEng, PlGF, and sFlt-1 measurements and corresponding ratios in the adverse and non-adverse group are displayed in Table 8.

## 3. Discussion

The pathophysiologic processes underlying preeclampsia and fetal growth restriction are complicated, multifactorial, and still unclear. In addition, the PE and FGR point to a common placental disorder and a common antiangiogenic state, but the development of maternal syndrome may require interaction with other factors to explain the disease in preeclampsia or perhaps the existence of some kind of protective agent against its appearance [11,12,13,14]. In reality, the balance between placental and maternal causations most likely varies among individuals across the spectrum of gestational age at clinical presentation. To date, it has not been sufficiently explained why the ischemic placenta in one disorder leads to growth restriction of the fetus and in the other to a maladaptation of systemic vasculature and inflammatory state, which constitute PE in the mother [15,16]. It was postulated that maternal PE arises from soluble factors released into the systemic circulation from the stressed placenta. Fetoplacental hypoxia is one of the consequences of placental impairment, which is a threat to the survival of the fetus.

The important findings of our study are that pregnancies with normotensive fetal growth restriction, isolated preeclampsia and preeclampsia with concurrent fetal growth restriction (iFGR, iPE and PE + FGR) share similar angiogenic profiles expressed in decreased PlGF and increased sFlt-1 and sEng maternal serum concentrations. However, differences between controls and investigated groups were most strongly pronounced in the PE with concurrent FGR group and by the ratio sFlt-1/PlGF (its median values in iPE and PE + FGR groups were 30 and 100 times higher than controls, respectively). The absolute highest values of the sFlt-1/PlGF ratio in preeclamptic cases with concurrent growth restriction of the fetus suggest accumulation of pathological changes reflected in extreme angiogenic disturbances in this group. These results support the hypothesis that although FGR and PE differ in their association with maternal disease, they share similar placental pathology responsible for changes in the angiogenic profile and thus, they share a common pathogenesis. Despite the heterogeneity in the phenotypic classification, disordered function of the maternal-fetal unit circulation is its hallmark feature, leading to the idea of considering PE and FGR as a single pathology named “ischemic placental syndrome” [15,16]. 

Placental ischemia is a key to their pathogenesis, but in preeclampsia it is the maternal systemic vasculature that is predominantly affected, while in FGR the major defect is limited to the placental and uterine vasculature presenting at risk fetal development and well-being. 

Nanjo et al. showed results consistent with our work, where in the PE complicated by FGR group, similar changes in sEng, sFlt-1 and PlGF serum levels were more markedly shown compared with PE alone [57]. Additionally, significant differences were observed between iFGR and controls with respect to sFlt-1 serum concentrations. sFlt-1 was the only substance analyzed that reached significantly higher values in both PE groups of patients, irrespective of concurrent FGR, when compared to healthy pregnant patients but did not differ between the iFGR group and controls.

Shibata et al. however found PlGF levels to be decreased in both PE and SGA, but sFlt-1 was increased only in the PE group [58]. However, authors included in the study isolated SGA, not precisely FGR patients.

This study is partially consistent with Alahakoon et al. who demonstrated similar, decreased PlGF profile in PE, PE + IUGR and isolated IUGR [59]. In their study sFlt-1 presented a significantly increased profile in all of these subgroups when compared to healthy controls.

Furthermore, significantly higher sEng serum values were observed compared to controls were identified in all investigated subgroups—iFGR, iPE and PE with simultaneous FGR. However, again (as in the case of PlGF and sFlt-1), the differences in sEng concentrations between the investigated subgroups were not significant. In addition, similar findings in pregnancies complicated by PE and SGA were presented by Levine et al. from the Karumanchi group, who demonstrated increased maternal levels of soluble endoglin in SGA patients from the second half of pregnancy onwards when compared to healthy controls, resulting in concentrations of this protein in the SGA group being as high as in term PE [26]. This study, however, did not involve the precise, updated definition of FGR based on placental insufficiency manifestations in Doppler studies but only fetuses with birth weight below the 10th percentile, which constitutes the definition of SGA. It was observed that sEng increased as sFlt-1 did in all investigated subgroups with either PE or FGR (*p* < 0.005, R = 0.57, 0.77 and 0.54 for the iFGR, iPE and PE + FGR groups, respectively), but there was no correlation between sFlt-1 and sEng in the control group. The sFlt-1*sEng/PlGF ratio was suggested to be predictive of PE [59,60]. There was a strong positive correlation in our study between sEng and sFlt-1 levels in the entire population (R = 0.69, *p* < 0.001). 

It was suggested that both entities, PE and FGR, share a common biological pathway, but they differ in the sequence of symptoms with attenuated sEng activity in isolated IUGR [60,61]. Thus, the maternal reaction in IUGR is supposed to be delayed. In this study, sEng did not present significant differences between groups, but we measured its concentration at the moment of prompt delivery when clinical decompensation of the mother or fetus was overt. However, the trend of sEng being elevated in normotensive FGR to a lesser extent than in PE, similar to sFlt-1, was visible (Figure 1A,B), and the PE with FGR group reached the highest levels of sEng, which is consistent with our previous work (mean values for PE + FGR, iPE and iFGR were 11.9, 10.5 and 9.9 ng/mL, respectively) [62]. 

It was found that pregnancies with fetal growth restriction, preeclampsia or HELLP, and preeclampsia or HELLP and fetal growth restriction showed significantly lower values of PlGF and a higher sFlt-1/PlGF ratio than control pregnancies in both groups with different gestational age: before and after 34 weeks (Table 9 and Table 10). The iFGR group presented clearly higher sFlt-1 concentrations than the control group, but the differences were not statistically significant (median values 33,590 vs. 8878 pg/mL, respectively). 

Crispi et al. found that in patients with early-onset PE and FGR, maternal PlGF levels were significantly lower and sFlt-1 levels were higher in all cases compared with healthy control subjects of similar gestational age and that all changes were more pronounced in PE than in FGR, but in the late-onset group, these differences were only seen in PE but not in the FGR group [39]. 

In our study, both PE groups with or without concomitant FGR presented significantly higher sFlt-1 in comparison to healthy controls. The median sFlt-1 values in isolated FGR were definitely lower than in iPE and PE + FGR, but higher than controls. The increased sFlt-1 values were more pronounced not only in PE with concomitant FGR, but also in early PE compared to the late form. Our interpretation is that it is sFlt-1 that is clearly directly linked to maternal disease and, considering generally lower values of sFlt-1 in late-onset disease when compared to early-onset, may be the factor responsible for never reaching the barrier needed to be crossed to trigger maternal response for the antiangiogenic state in iFGR patients. This is in line with previous studies by Levine, Chaiworapongsa and Koga that claimed that circulating sFlt-1 concentrations are increased in women with an established diagnosis of PE and may begin to increase weeks before the onset of clinical symptoms [23,24,34]. Thus, sFlt-1 appears to be the central mediator of preeclampsia, but it is not is not indifferent to the development of FGR.

Although sFlt-1 plays an important role in the pathogenesis of PE, it is unlikely that sFlt-1 alone governs disease onset. In our comparison, sFlt-1 or any of its ratios did not differ significantly between iFGR and PE pregnancies with or without concurrent FGR. This led us to conclude that other circumstances and multiple factor interactions are necessary to progress from angiogenic imbalance caused by increased sFlt-1 to maternal disease.

It has been hypothesized that the development of maternal endothelial dysfunction in preeclampsia occurs after a certain threshold of imbalance is reached. Further, the women with predisposing conditions may develop PE earlier, with milder angiogenic imbalance. The cardiovascular predisposition of some pregnant women to PE is a currently investigated hypothesis on its etiology [63]. The hypothesis assumes that any placental dysfunction during pregnancy is secondary to underlying subclinical defects in women’s cardiovascular system [64]. It is becoming increasingly evident that pregnancy presents a significant strain on the maternal cardiovascular system and metabolism, and in women with evidence of significant maladaptation, preeclampsia is the clinical phenotype. For this reason, pregnancy has been described as a ‘stress test’ that unmasks women who have poor cardiovascular reserve or dysfunction [65]. Individual women with prepregnancy risk of developing cardiovascular syndromes, such as obesity, under this pregnancy stress test will develop PE. FGR develops in the absence of maternal constitutional predisposition, whereas PE develops when mediators of placental stress, such as sFlt-1, provoke sufficiently severe and prolonged endothelial insult and interact with chronic hypertension, renal disease, thrombophilia, insulin resistance/diabetes and obesity [66]. In women with long-term cardiovascular risk factors, even physiological changes at the end of pregnancy and especially late-onset placental disease, which is more commonly associated with less or no placental damage, may induce endothelial dysfunction leading to PE [56,67].

Thus, in some women the degree of placental dysfunction may be high, but with low predisposition to maternal inflammation the final result is FGR without maternal complications. Some women may also have a resistant-type of vasculature that may therefore take longer to develop clinical signs and symptoms in response to placental-shed inflammatory factors. Clinically, these pregnancies can be rescued by iatrogenic delivery when severe FGR is detected on time, before they develop the overt stage of PE. The removal of dysfunctional placenta protects them from symptomatic PE. In these women the threshold needed to develop maternal disease is never reached.

Therefore, it is possible that very high maternal soluble Flt-1 levels are required for preeclampsia to develop. The highest sFlt-1 values in both PE groups, with or without FGR, suggest that this substance is of essential significance for the development of PE. It may be one of the key substances needed for reaching the threshold for PE to develop as a maternal manifestation of ischemic placental disease. Nevertheless, sFlt-1 serum values overlap in many cases in PE and FGR, which Figure 1C illustrates very well. The threshold to be crossed in mothers might be wide range, as women developing PE are a heterogeneous group with different health backgrounds, predispositions and tolerance to physiological and pathological changes taking place in their organisms during pregnancy.

Herraiz et al. proposed a very interesting term of “antiangiogenic continuum” as a proxy of deteriorating placental function, which reflects an observed tendency of the sFlt-1/PlGF ratio to increase from iFGR, toward iPE and finally PE with FGR [14]. This gradual, significant growth is also visible in our study. It is mostly discernible for median concentration of sFlt-1 in each subgroup: starts with the lowest values in iFGR, reaches median results in iPE and finishes with the highest median value in PE + FGR, which is visualized on previous Figure 1B. It seems that this might be interpreted as a more severe placental involvement during FGR in the course of PE as well as at early-onset disease, as these two mechanisms are synergistic: these two entities, FGR and PE converge more frequently in pregnancies at younger gestational age which is attributed to our PE + FGR subgroup.

The division of our relatively small group of patients with iFGR into the early and late subgroups showed significant differences in the level of PlGF concentration, as well as sFlt-1, and thus also their ratio. There was no significant difference in sEng concentration between the early and late FGR groups. For early and late PE, we noted significant differences only in terms of serum concentration of PlGF (data presented in Table 9 and Table 10).

The change in the equilibrium in serum between pro- (PlGF) and antiangiogenic (sFlt-1) factors as well as the level of the resulting sFlt-1/PlGF ratio seems to play a role not only in the diagnosis of PE but also as a predictor of adverse outcomes and gestational time remaining until delivery. Strongly elevated sFlt-1/PlGF ratios in early and late-onset PE have been observed to be correlated with the necessity of delivering the fetus within 48 h and the need for immediate and careful fetal monitoring. The ratio could be helpful in decision-making when inpatient monitoring and fetal lung maturation are being considered.

FGR is also an obstetric complication that presents huge challenges in terms of carrying out diagnostic tests and monitoring the fetus. The complexity of diagnostic criteria is a representative of how complicated the syndrome is. The ultrasound is of primary usefulness here, where apart from assessing the fetal weight, it is used to measure Doppler flows in the uterine, umbilical and middle cerebral arteries [68]. However, the results of Doppler studies may be somewhat delayed in relation to the onset of insufficiency, as some studies have shown quite severe ischemic placental lesions despite the diagnosis of normal UA flows [69]. It seems that combined with ultrasound (fetal biometry, feto-maternal Doppler studies), the sFlt-1/PlGF ratio appears to be useful as a supplementary criterion not only for the detection of FGR but also for the prediction of the time-to-delivery interval and associated adverse outcomes in isolated FGR cases [70,71,72]. In our research, we observed a positive correlation between sFlt-1/PlGF ratio values and uterine artery PI (R = 0.8, *p* < 0.00005), umbilical artery PI (R = 0.52, *p* < 0.00005) and RI (R = 0.47, *p* < 0.00005) in the entire study population using Spearman’s correlation analysis. These correlations were also observed separately for iFGR and both PE groups together (including iPE with PE + FGR).

In putting our work into a clinical, practical perspective, we compared the most widely analyzed ratio of sFlt-1 to PlGF in investigated subgroups according to gestational age with preliminary cutoffs with suggested clinical application in PE syndrome. These preestablished cutoffs of ratio values were: 38, 85, 110, 201 and 655 and their recommended interpretation and clinical application in PE diagnosis and prognosis are shown in Table 11 [55,56].

In the control group, the median and mean values of the sFlt-1/PlGF ratio were 10 and 18, respectively, and the vast majority of patients (17/20, 85%) belonged to the low ratio group. This is quite expected, as these patients were totally healthy during the entire pregnancy, with no underlying medical conditions. Currently, the cutoff level of ≤38 is widely accepted for ruling out PE in patients with suspicion of the disease, and has already been implemented in some European countries (such as Spain, Italy, the UK, France and Germany) [56,73]. There was a small representative of the control group (3/20, 15%) in the intermediate ratio group but none of these patients exceeded the gestational-wide threshold of 85 for PE diagnosis [74]. Notably, all of these patients were at advanced gestational age (mean ratio value is 50 and mean gestational age is 37.3 weeks). The explanation for the finding that all patients from the control group that crossed the threshold of 38 are at term is the fact that advanced, close to term but eventually normal pregnancy exhibit changes in biomarkers sFlt-1 and PlGF similar to those in PE—the ratio increases. In early healthy pregnancy, proangiogenic substances prevail, whereas PE is considered to be an antiangiogenic state. However, the trend that aims to reach this antiangiogenic state can be observed in advancing physiological pregnancies as well. In our study we noted higher concentrations of sFlt-1 and sEng, and lower PlGF values in patients > 34 weeks in the control group in comparison to those at lower gestational age, <34 weeks (*p* < 0.005 for PlGF and <0.05 for sEng and sFlt-1). As the syncytiotrophoblast (STB) is the main placental source of soluble fms-like tyrosine kinase 1, PlGF and sEng and secretes these factors predominantly into the maternal circulation where they impact maternal vascular adaptation to pregnancy, we can interpret the increasing sFlt-1/PlGF ratio at term as an indicator of accumulating STB stress, expanding with advancing gestation [31,75]. STB pathology, at or after term (for example focal STB necrosis or syncytial knots) demonstrates this stress with or without the occurrence of PE [76], which is supported in our study although the number of women is very low. It should be emphasized that that postterm pregnancies are also complicated by increased rates of PE and more strikingly, eclampsia [77]. It also implies that all pregnant women may be somehow destined to suffer from PE, but spontaneous or induced delivery averts this outcome in most instances [78]. 

In each study cohort, 87%, 79% and 64% for PE + FGR, iPE and iFGR respectively, met the PE diagnostic threshold of 85. Furthermore, most of the patients from study subgroups reach extremely high, defined above 655 or 201 cutoffs for early and late onset PE [74,77]. These patients in our study delivered within the following 48 h. When the preliminary cutoff value of 85 [75] for PE diagnosis for gestation-wide was applied, all investigated subgroups—iFGR, iPE, PE + FGR had a mean sFlt-1/PlGF ratio above that cutoff meeting the diagnostic criteria of PE even with isolated FGR and no overt maternal disease. The mean/median values for the PE + FGR, iPE, and iFGR groups were 2577/1072, 1181/314, and 408/218, respectively. These patients in our study delivered within the following 48 h. At delivery, most patients with isolated FGR without maternal disease reach the diagnostic criteria for PE of a ratio of more than or equal to 85. In our cohort, all patients in this high-ratio group had only early-onset PE.

It was observed that a sFlt-1/PlGF ratio > 655 at diagnosis was associated with a 5-fold increased risk of delivery in ≤48 h [79]. In our study, all of the patients delivered within the next 48 h, and 67% of the entire study group was allocated to the very high sFlt-1/PlGF ratio cohort. 

The blood samples were collected within the last 48 h prior to delivery, when disturbances in angiogenic substances are maximized at the most critical point of pregnancy, when aggravating the fetal state in utero is no longer acceptable and evacuation is necessary, disturbances in the angiogenic profile reach levels as high as those in most serious PE cases. It was expected that the angiogenic and antiangiogenic factors would be dysregulated at this point, but to a less dramatic extent. 

In both preeclampsia and fetal growth restriction, placental dysfunction with fetoplacental hypoxia has been suggested as a common underlying condition that may provide an explanation for their strong associations: therefore, the sFlt-1/PlGF ratio may be useful for predicting adverse outcomes not only in PE, but also in women with FGR [80,81]. 

Table 7 lists the occurrence of adverse outcomes by mortality and morbidity events that occurred in our patient group. In our study, most events occurred in the PE + FGR group of patients, which was also the group with the statistically lowest gestational age. In the FGR group, one preterm abruptio placentae and one intrauterine death occurred.

It was discovered that the combination of both entities, fetal growth restriction age and preeclampsia are exceptionally strongly associated with the elevated sFlt-1 to PlGF ratio. 

Additionally, these findings and data from literature seem support both entities e.g FGR and preeclampsia should be considered as different manifestations of the same placental disease. It may be also suggested that an extreme angiogenic imbalance reflects a status of severe placental disease, in which there is a high risk of developing a broad spectrum of maternal and fetal complications that must be assessed comprehensively, regardless of whether a reliable diagnosis of preeclampsia has been reached.

## 4. Materials and Methods

### 4.1. Patients

A prospective cross-sectional case control study was conducted on 77 patients aged 20–41 years, between 24 and 41 weeks of gestation. Eligible cases were live singleton pregnancies with a diagnosis of new-onset PE and/or FGR hospitalized in a Polish tertiary referral hospital. In the entire study group, 75% (43/57) were PE patients (with or without concurrent FGR) and 25% (14/57) were isolated FGR cases. The patients were recruited into three subgroups: PE with concurrent FGR (*n* = 22), isolated PE (*n* = 21), and isolated normotensive FGR (*n* = 14). The entire study group was pairwise matched by gestational age with healthy control pregnancies (*n* = 20). The angiogenic disturbances were also checked in the group of adverse (*n* = 23) and no adverse outcomes (*n* = 34) among the entire cohort levels were observed at the most critical moment of pregnancy, right before delivery. The samples were collected immediately before delivery due to maternal or fetal clinical aggravation where the differences in terms of angiogenic imbalance are maximized.

Additionally, very precise inclusion criteria were used for the study groups, according to the newest, updated definitions for the diagnosis of both entities formed by ISSHP for PE in 2018 [82] and by FIGO (based on a Delphi consensus by Gordijn, reached in 2016) for FGR in 2021 [83,84]. The aim was to assess whether disordered angiogenesis markers and their ratios differed between the study groups and the control group, and among study subgroups (isolated PE, isolated FGR, and PE with simultaneous FGR) at the moment of indicated prompt delivery due to maternal or fetal aggravation.

The PE was defined according to the criteria applied in 2018 by the International Society for the Study of Hypertension in Pregnancy (ISSHP) [82] Group: the new onset of hypertension (BP ≥ 140 mm Hg systolic or ≥90 mm Hg diastolic) on two or more consecutive occasions accompanied by new-onset proteinuria (>0.3 g/24-h in 24-h urine collection) or, in the absence of proteinuria, another maternal organ or uteroplacental dysfunction. The detailed signs and symptoms required for PE diagnosis by ISSHP are listed in Table 9. All the PE patients in our study met the diagnostic criteria independently of developing uteroplacental dysfunction manifested by growth restriction, i.e., apart from gestational hypertension they presented signs of maternal organ dysfunction or proteinuria, despite the concurrent FGR in the PE plus FGR group. We included only preeclampsia patients diagnosed after 20 weeks of gestation and excluded patients with multiple gestations, chronic hypertension, nephrotic syndrome diagnosed postpartum, diabetes, other metabolic disorders, or autoimmune disease. The absence of fetal structural or chromosome abnormalities was also required.

Fetal growth restriction was diagnosed according to the Delphi consensus-based definition for placenta-mediated FGR published by Gordijn et al. in 2016 [83], recognized recently in 2021 by the FIGO initiative on fetal growth, which uses a combination of measures of fetal size percentile and Doppler abnormalities for early and late FGR. Data presented in Table 12. According to the consensus definition of FGR, we defined FGR as an estimated fetal weight (EFW) < 3rd percentile or EFW < 10th percentile in combination with at least one of the following Doppler abnormalities: umbilical artery (UA) pulsatility index(PI) > 95th percentile, middle cerebral artery (MCA) PI < 5th percentile, cerebroplacental ratio (CPR) < 5th percentile and/or a mean uterine artery (mUtA) PI > 95th percentile. The cases with a birth weight exceeding the 10th percentile for gestational age, and those with placental or umbilical cord abnormalities, anatomical malformations or suspicion of any genetic defect were excluded from the FGR group. 

The healthy normal pregnancies were recruited in our outpatient department to gestationally match the pathologic cases. Inclusion criteria were noncomplicated singleton pregnancy with absence of labor at the time of venipuncture. The patients were verified postdelivery, and the control group included only women with delivery of a term (>37 weeks) infant whose birth weight was between the 10th and 90th percentile for gestational age and no medical, obstetrical nor surgical complications during the entire gestation.

Additionally, in giving this work clinical relevance, the sFlt-1 to PlGF ratio at delivery was then classified according to current recommendations for its practical use in risk stratification in suspected or diagnosed PE. These preestablished cutoffs of ratio values were: 38, 85, 110, 201 and 655 and their recommended interpretation and clinical application in PE diagnosis and prognosis are shown in Table 11. The values of the sFlt-1 to PlGF ratio in each cohort were then allocated to low, intermediate, high or very high-risk groups according to up-to-date expert opinion on the practical use of angiogenic markers.

Finally, there was a comparison between angiogenic biomarker concentrations and the adverse and non-adverse pregnancy outcome groups. The maternal and fetal adverse outcomes were based on the CHIPS (Control of Hypertension in Pregnancy Study), PIERS (Preeclampsia Integrated Estimate of RiSk) and PREP studies, where components of the outcome were derived through an iterative Delphi consensus process [85,86,87]. An independent panel of experts ranked the outcomes for their importance to clinical practice [88]. Data presented in Table 13. 

The table is based on work of: Herraiz I, Llurba E, Verlohren S, Galindo A. Spanish Group for the study of angiogenic markers in preeclampsia. The update on the diagnosis and prognosis of preeclampsia with the aid of the sFlt-1/PlGF ratio in singleton pregnancies. FetalDiagnTher 2018;43:81–89 [56] and Stepan H, Herraiz I, Schlembach D, Verlohren S, Brennecke S, Chantraine F, Klein E, Lapaire O, Llurba E, Ramoni A, Vatish M, Wertaschnigg D, Galindo A: Implementation of the sFlt-1/PlGF ratio for prediction and diagnosis of preeclampsia in singleton pregnancy: implications for clinical practice. Ultrasound Obstet Gynecol 2015;45:241–246 [55].

### 4.2. Methods

The patients who had been referred to the tertiary-level academic unit for further treatment because of preeclampsia and/or fetal growth restriction without any signs of labor were offered participation in this study.

In all pathologic pregnancies, samples were collected at the moment of clinical detrioration and indication for prompt delivery due to maternal or fetal clinical decompensation, when the differences in terms of angiogenic imbalance were expected to be maximized. As a result, all the samples were collected within the last 48 h of pregnancy duration. The angiogenic substance levels were not known at this point, and they were not included as part of patient management.

The serum samples were collected according to a common standard operating procedure at our center. Further, the venous blood was drawn by venipuncture in tubes without anticoagulant directly from the patient. In the case of immediate cesarean section, blood was collected for routine blood tests and the remaining serum samples were obtained from the hospital laboratory. Once collected, the blood samples were centrifuged at 2000× *g*, and the serum was separated, aliquoted and stored at −80 °C for later analysis. The levels of PlGF, sEng and sFlt-1 in maternal serum were measured by the corresponding sandwich enzyme immunoassay technique kits (R&D Systems Europe Ltd., Abingdon, UK) according to the manufacturer’s instructions. The sEng, sFlt-1 and PIGF concentrations of each sample were determined in parallel.

The fetal measurements and Doppler studies were performed at the Clinical Department of Obstetrics and Perinatology using curvilinear transabdominal probe and a Voluson E10 device (GE Medical Systems). The ultrasound was carried out by senior obstetricians upon routine conditions and guidelines. Biometry was performed by measuring the abdominal circumference (AC), biparietal diameter (BPD), head circumference (HC) and femur length (FL). The fetal weight and the weight percentile were calculated using the Hadlock curves [53]. The following Doppler parameters were measured: PI of the uterine arteries (UtA), PI of the umbilical artery (UA), PI of the middle cerebral artery (MCA), PI and the cerebroplacental ratio (CPR) as the ratio between MCA PI and UA PI and calculations were performed according to up-to-date reference ranges [89,90].

The clinical data were ascertained prospectively and included age, height, weight, verified gestational age, past medical and obstetrical history, as well as clinical information including blood pressure, medication, and laboratory test values.

All data were entered into a database in a real time. 

Early perinatal outcomes including birthweight and Apgar score were recorded. A total of 77 pregnant women out of initially involved 88 fulfilled the criteria for their inclusion in the study.

The study protocol was approved by the Medical University of Lublin Ethics Committee (KE-0254/258/2016). Written consent was obtained from all participants in the study.

### 4.3. Statistical Analysis

The statistical comparisons between healthy control participants and the entire study group as well as comparisons between study subgroups: PE + FGR, iPE, iFGR were performed using Statistica 13.1 by StatSoft. The analysis was conducted with Kruskal-Wallis test with Bonferroni’s adjustment and analysis of variance ANOVA with post hoc RIR Tukey test when possible. Box plots were generated to represent the values of analytes and their ratios in different subgroups. The previously described gestational phase-specific cutoff values of the sFlt-1/PlGF ratio for PE diagnosis and short-term outcome prediction were applied for different outcome groups [55,56]. The criterion for statistical significance was *p* < 0.05.

## 5. Conclusions

A strong subsequent antiangiogenic activity may be interpreted as a response to hypoxia, and high levels of soluble Flt-1 and soluble endoglin may be markers of this activity. In addition, a positive correlation between sFlt-1 and sEng suggests that antiangiogenic pattern in VEGFs and TGF-beta pathways provides a coordinated reaction to fetoplacental hypoxia.

The fact that PlGF, sFlt-1 and sEng concentrations at delivery as well as their ratios overlap in different manifestations of placental ischemia—FGR and PE, and in their early and late forms constitute proof that mechanisms behind their development finally lead to similar placental responses and cannot be separated. Further, multiple factors must interact and this additive mechanism needs time to give manifestation sooner or later. The complex interactions described above highlight, that molecules such as PlGF, sFlt-1 and sEng cannot be considered in isolation but in combination with different substances that reflect maternal systemic vasculature function. The potential application of biomarkers of angiogenesis in both FGR and PE to detect or predict patients at risk of adverse outcomes would be of value to improve risk stratification strategies to avoid serious complications of ischemic placental disease. 

Additionally, from a practical point of view, the fact that at the end of pathological pregnancy complicated by FGR and PE, disturbances in angiogenic profile in maternal blood reach similar levels, points to their possible use in FGR as they ultimately start to be applicable in PE management. 

Finally, this work supports the hypothesis that the determination of the sFlt-1/PlGF ratio in patients with FGR might also have value for counseling, clinical supervision, and risk anticipation as it has just started to be applied in PE management. This necessitates further research and a better integration of biomarkers in the diagnostic work-up.

## 6. Strengths and Limitations

The cohort was constructed with a very precise, up-to-date criteria for PE and FGR. The FGR diagnosis was confirmed for each newborn after delivery and patients not meeting the criteria were excluded. In an attempt to decrease the likelihood of false-positive and false-negative diagnosis of FGR, the consensus definition was based on a combination of measures of fetal size (fetal weight estimation and abdominal circumference) and abnormal Doppler findings in the umbilical, uterine, and middle cerebral arteries, as described in Table 10. In our study blood samples were collected directly before the delivery, which was performed within the next 48 h from the moment of venipuncture, due to maternal or fetal clinical decompensation, when the differences in terms of angiogenic imbalance are maximized, while many studies concentrate on prediction or moment of diagnosis in suspected PE cases, when changes in angiogenic profile are less pronounced in late- onset disease. It was only found in one work that was focused on angiogenic factor levels within one week prior to delivery in similar cohorts—iPE, iFGR and those entities combined together by Nanjo et al. The results regarding concentrations of PlGF, sFlt-1 and sEng in investigated subgroups were exactly the same as those presented by Japanese authors. This study has certain limitations. It was done in a single center and it is of relatively small size. This is why we did not separate early-onset from late-onset patients in study cohorts for deeper analysis. Although there was an interested in all preeclampsia-related adverse outcomes, rare but meaningful adverse outcomes such as pulmonary edema, DIC or maternal death, were not observed in the patients.

## Figures and Tables

**Figure 1 ijms-24-00972-f001:**
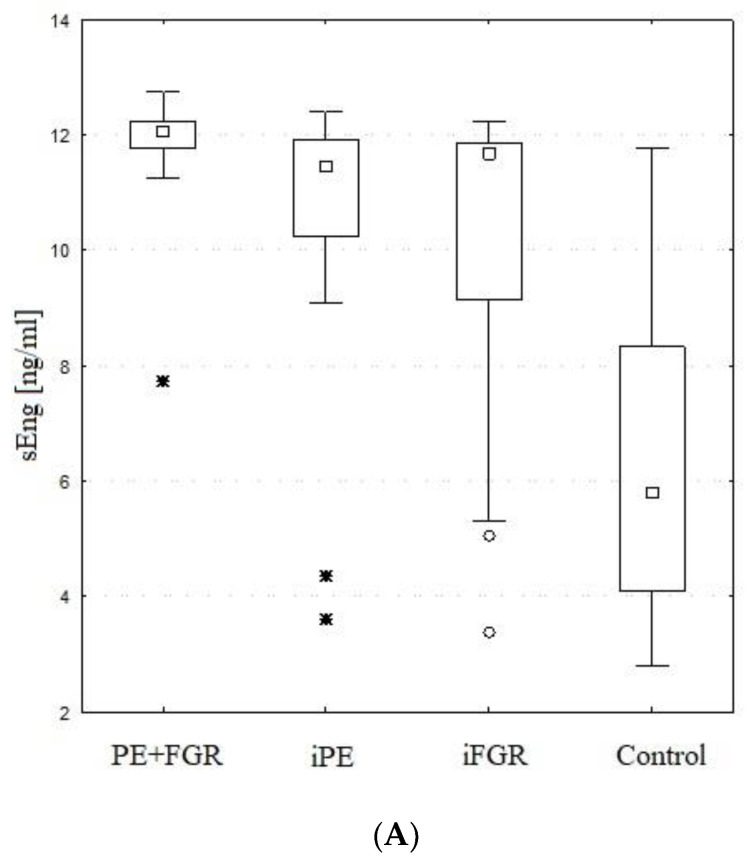
(**A**–**E**) Box and whisker plots for serum concentrations of sEng, sFlt-1, PlGF and sFlt-1/PlGF and sFlt-1*sEng/PlGF ratios in study groups: PE + FGR (preeclampsia with concurrent FGR), iPE (isolated PE), iFGR (isolated, normotensive FGR) and healthy control patients. Boxes indicate interquartile range; whiskers indicate range; square bars indicate median, asterisks indicate extreme values.

**Table 1 ijms-24-00972-t001:** Basic characteristics and statistical analysis of the study groups (based on results of the Kruskal-Wallis test and analysis of variance ANOVA with a post hoc RIR Tukey test).

	PE + FGR		iPE		iFGR		Control			
	I		II		III		IV			
Parameter	Median	Q1–Q3	Median	Q1–Q3	Median	Q1–Q3	Median	Q1–Q3	*p*-Value	Differences
Gravidity	1	1–2	1	1–2	2	2–3	2	1.5–3	ns	-
Parity	1	1–2	1	1–2	2	1–2	2	1–2	ns	-
Gestation age (weeks *)	32	28–34	35	33–37	35	33–37	34	31–37	ns	I < II
Age (years)	29	27–35	30	27–34	33	30–37	29	28–37	ns	-
Height (cm)	167	160–170	164	160–168	167	164–171	165	164–168	ns	-
Weight (kg)	72	66–89	80	72–92	70	67–79	78	68–89	ns	-
**I trimester**										
Weight (kg)	63	56–76	67	61–75	60	58–69	71	65–75	ns	-
BMI	24	20–26	25	23–28	23	21–26	28	24–28	ns	-
MAP	95	88–97	93	83–96	88	87–92	79	75–91	<0.05	I, I + II > IV
DBP (mmHg)	80	76–82	77	68–80	75	70–78	67	62–76	<0.05	I, I + II > IV
SBP (mmHg)	120	110–130	126	115–129	119	116–124	111	100–120	<0.05	I + II > IV

*p*-value reflecting statistically significant differences between the study groups and the control group. ns refers to the nonsignificant differences between the control and investigated groups. Q1–Q3 refers to the interquartile range; BMI refers to the body mass index; MAP is the arterial pressure; DBP refers to the diastolic blood pressure; SBP refers to the systolic blood pressure; Gestation age (weeks *) refers to the gestational age at the moment of venipuncture.

**Table 2 ijms-24-00972-t002:** Clinical results at the time of evaluation before delivery and statistical analysis of the study groups (based on the results of the Kruskal-Wallis test and analysis of variance ANOVA with a post hoc RIR Tukey test).

	PE + FGR		iPE		iFGR		Control			
	I		II		III		IV			
Parameter	Median	Q1–Q3	Median	Q1–Q3	Median	Q1–Q3	Median	Q1–Q3	*p*-Value	Groups
SBP max	170	156–178	156	150–165	127	115–134	114	104–122	<0.005	I > II > III > IV
DBP max	104	102–111	98	95–105	81.5	76–84	66	62–75	<0.00005	I,II > IV
MAP	128	121–131	117.3	114–127	95.7	92–98.7	84	76–89	<0.00005	I > II > III > IV
Proteinuria (mg/24 h)	1438	547–3483	668	295–1981	170	138–192	0	0	<0.00005	I,II > IV
Total protein (g/dL)	6.05	5.8–6.3	5.8	5.6–6.3	6.35	6–6.7	6.0	5.5–6.2	ns	-
Fibrinogen (g/L)	4.4	3.7–5	5.2	4.3–5.6	4.7	4–5.2	3.85	3.5–4.2	<0.05	II > IV
INR	0.9	0.86–0.93	0.9	0.9–0.98	0.92	0.9–0.97	1.0	1–1	<0.00005	I,II,III < IV
PT Index(%)	109	105–114	106	100–110	105.3	102–110	98	96–100	<0.005	I,II,III > IV
PT (s)	10	10.4–9.6	10.3	10.9–9.9	10.4	10.7–9.9	11.1	1.1–10.9	<0.005	I,II,III < IV
APTT (s)	28.8	26.6–29.6	26.5	25.6–27.3	27.4	26.2–29.6	26.9	25.9–29.4	ns	-
D-dimers (ng/mL)	1277	1050–1800	1362	1140–1731	1306	951–1670	1359	923–2389	ns	-
WBC (×10^9^/L)	10.3	8.8–11.4	9.8	9–12.4	9.4	7.5–11.4	9.05	8.4–10.9	ns	-
RBC (×10^12^/L)	4.16	3.95–4.34	4.0	3.8–4.3	4.2	4–4.3	4.0	3.7–4.2	ns	-
Hb	12.5	11.8–13.5	12.2	11.4–13.2	12.6	12–13.1	12.25	11.5–12.7	ns	-
HCT (%)	37	34.3–38.8	34.9	33.7–38.5	36.8	35.6–37.9	35.3	33.9–37.2	ns	-
PLT (×10^9^/L)	192	147–220	189	144–213	207	178–253	220	179–263	ns	-
ALT (U/L)	28.5	21–76	24.5	18–46	16	14–28	17	12–18	<0.005	I,II > IV
AST (U/L)	42	30–69	30	25–41	23	20–28	20	15–21	<0.00005	I,II > IV
Creatinine (mg/dL)	0.7	0.6–0.8	0.7	0.6–0.8	0.65	0.5–0.7	0.5	0.5–0.6	<0.05	I, I + II > IV
UA (mg/dL)	7.2	6.7–8.2	6.5	5.5–7.2	5.5	3.95–6.6	4.2	3.2–4.6	<0.00005	I,II > IV
Urea	31	26.3–42.1	24.5	20–35	19	19–26	15.6	13.7–17	<0.00005	I,II > IV

*p*-value reflecting statistically significant differences between the study groups and the control group. ns refers to the nonsignificant differences between the control and investigated groups. Q1–Q3 refers to the interquartile range; MAP refers to the mean arterial pressure; DBP max refers to the maximum value of diastolic blood pressure; SBP max refers to the maximum value of systolic blood pressure; INR refers to the international normalized ratio; PT refers to the prothrombin time; APTT refers to the activated partial thromboplastin time; WBC refers to the white blood cell count; RBC refers to the red blood cell count; Hb refers to the hemoglobin concentration; HCT refers to the hematocrit; PLT refers to the platelet count; ALT refers to the alanine transaminase; AST refers to the aspartate transaminase; UA refers to uric acid.

**Table 3 ijms-24-00972-t003:** Characteristics and statistical analysis of ultrasound, including Doppler parameters, in the study groups (based on results of the Kruskal-Wallis test and analysis of variance ANOVA with a post hoc RIR Tukey test).

	PE + FGR		iPE		iFGR		Control			
	I		II		III		IV			
Parameter	Median	Q1–Q3	Median	Q1–Q3	Median	Q1–Q3	Median	Q1–Q3	*p*	Differences
UtPI mean	1.6	1.4–2.0	1.0	1–1	1.6	1.3–2	0.7	0.65–0.75	<0.0005	I,III > IV
Ut PI pc	100	99–100	91	85–96	100	100–100	52	42–62	<0.0005	I,III > IV
UA PI	1.3	1–2.42	1.0	0.9–1	1.2	1–1.7	0.8	0.65–0.87	<0.00005	I,II,III > IV
UA PI pc	84	53–100	73	59–82	96	70–100	11	3–46	<0.00005	I,II,III > IV
UA RI	0.8	0.6–1.0	0.6	0.6–0.7	0.7	0.6–0.8	0.6	0.49–0.58	<0.0001	I,III > IV
MCA PI	1.2	1.1–1.7	1.6	1.5–2	1.3	1.3–1.5	1.6	1.2–1.7	<0.05	I < II
MCA PI pc	1	1–23	32	12–46	3	1–15	21	8–35	<0.05	I < II
CPR	1.2	0.6–1.6	1.6	1.5–1.8	1	0.7–1.6	2.0	1.7–2.2	<0.0005	I,III < IV
CPR pc	1	1–16	19	6–35	1	1–13	51	30–83	<0.0005	I,III < IV
AFI	6.5	3–10	10	8–14	10	7.5–11	11	9–14	<0.05	I < II
EFW	1326	708–1714	2760	2167–3173	1915	1464–2255	2607	1773–3351	<0.00005	I < IV
EFW pc	1	1–2	56	33–83	2	1–5	64	43–87	<0.00005	I,III < IV
AC	242	217–260	323	299–336	281	245–286	304	260–342	<0.0005	I < IV
AC pc	1	1–5	58	48–74	4	1–7	57	40–81	<0.00005	I,III < IV

*p*-value reflecting statistically significant differences between the study groups and the control group. ns—nonsignificant differences between the control and investigated groups. Q1–Q3—interquartile range. Ut PI (uterine artery pulsatility index); Ut PI pc (centile of uterine artery pulsatility index); UA PI (umbilical artery pulsatility index); UA PI pc (centile of umbilical artery pulsatility index); UA RI (umbilical artery resistance index); MCA PI (middle cerebral artery pulsatility index); CPR (cerebroplacental ratio); CPR pc (centile of cerebroplacental ratio); AFI (amniotic fluid index); EFW (estimated fetal weight); EFW pc (centile of estimated fetal weight); AC (abdominal circumference); AC pc (centile of abdominal circumference).

**Table 4 ijms-24-00972-t004:** Characteristics and statistical analysis of neonatal outcomes of the study groups (based on results of the Kruskal-Wallis test and analysis of variance ANOVA with a post hoc RIR Tukey test).

	PE + FGR		iPE		iFGR		Control			
	I		II		III		IV			
Parameter	Median	Q1–Q3	Median	Q1–Q3	Median	Q1–Q3	Median	Q1–Q3	*p*	Differences
Gestational age at birth (weeks)	32	28–34	35	33–37	35	33–37	34	37–39	ns	-
Birth weight (g)	1370	680–1700	2500	1980–2980	1985	1480–2320	3340	3170–3520	<0.00005	I,II,III < IV
Birth percentile Hadlock * 1991	1	1–2	33	18–81	3.5	1–8	61	37–77	<0.00005	I,III < IV
Birth percentile Akolekar ** 2018	1	1–1	46	22–86	3	1–7	70	47–87	<0.00005	I,III < IV
Apgar 1 minute	7	6–8	8	7–10	8	7–10	10	9–10	<0.0005	I,II < IV
Apgar 5 min	7.5	6–9	9	8–10	8	8–10	10	9.5–10	<0.00005	I < IV

*p*-value reflecting statistically significant differences between the study groups and the control group. Ns refers to the nonsignificant differences between controls and investigated groups. Q1–Q3 refers to interquartile range. * Hadlock FP, et al., In utero analysis of fetal growth: a sonographic weight standard. Radiology. 1991 Oct;181(1):129–33 [53]. ** Nicolaides KH, Wright D, Syngelaki A, Wright A, Akolekar R. Fetal Medicine Foundation fetal and neonatal population weight charts. Ultrasound ObstetGynecol 2018; doi:10.1002/uog.19073 [54].

**Table 5 ijms-24-00972-t005:** Distributions of the values of sFlt-1, sEng, PlGF and the sFlt-1/PlGF and sFlt-1*sEng/PlGF ratios in women with isolated PE, isolated FGR, combined PE and FGR and in the control group (based on the results of the Kruskal-Wallis test with Bonferroni adjustment).

	Group	M	Me	Q1	Q3	SD	H, *p*
sEng [ng/mL]	(I) PE + FGR	11.9	12.1	11.8	12.2	1.0	H = 34.598
	(II) iPE	10.5	11.5	10.2	11.9	2.4	*p* < 0.001
	(III) iFGR	9.9	11.7	9.2	11.9	3.0	I > IV, II > IV, III > IV
	(IV) Control	6.4	5.8	4.1	8.3	2.9	
PIGF [pg/mL]	(I) PE + FGR	72	42	22	113	62	H = 42.603
	(II) iPE	149	142	27	227	118	*p* < 0.001
	(III) iFGR	216	154	117	221	261	I > IV, II > IV, III > IV
	(IV) Control	851	769	444	1248	480	
sFlt-1 [pg/mL]	(I) PE + FGR	129,263	115,702	14,981	221,278	123,234	H = 14.510
	(II) iPE	87,234	76,345	8614	133,888	99,327	*p* = 0.002
	(III) iFGR	51,193	33,590	13,871	66,994	49,647	I > IV, II > IV
	(IV) Control	9787	8878	5574	10,809	6416	
RATIO sFlt-1/PlGF	(I) PE + FGR	2577	1072	250	2833	4638	H = 46.100
	(II) iPE	1181	314	143	547	3567	*p* < 0.001
	(III) iFGR	408	219	81	846	438	I > IV, II > IV, III > IV
	(IV) Control	18	10	5	24	17	
RATIO sFlt-1*sEng/PlGF	(I) PE + FGR	31,549	12,859	3013	33,345	56,617	H = 44.965
	(II) iPE	13,832	3251	1492	6418	41,999	*p* < 0.001
	(III) iFGR	4727	2587	646	10,109	5350	I > IV, II > IV, III > IV
	(IV) Control	142	53	24	187	191	

*p*-value statistically significant differences between the study groups and the control group. M—mean value, Me—median value, Q1–Q3—interquartile range.

**Table 6 ijms-24-00972-t006:** Distribution of the sFlt-1/PlGF ratio in each study subgroup in the low, intermediate, high or very high risk groups according to the currently recomended cutoff values.

Ratio sFlt-1/PlGF		PE + FGR (*n* = 22)	iPE (*n* = 21)	iFGR (*n* = 14)	Control (*n* = 20)
Mean/Median		2577/1072	1181/314	408/218	18/10
Low		5%	0	0	85%
<38		1 (5%)	0	1 (7%)	17 (85%)
Intermediate		9%	24%	29%	15%
38–85	<34 w	2 (9%)	1 (5%)	0	0
38–110	>34 w	0	4 (19%)	4 (29%)	3 (15%)
High		14%	14%	0	0
>85	<34 w	3 (14%)	3 (14%)	0	0
>110	>34 w	0	0	0	0
Very high		16 (73%)	13 (65%)	9 (64%)	0
>655	<34 w	10 (45%)	3 (14%)	4 (29%)	0
>201	>34 w	6 (27%)	10 (48%)	5 (36%)	0

Cutoff values are based on the following: (1) Stepan H, Herraiz I, Schlembach D, Verlohren S, Brennecke S, Chantraine F, Klein E, Lapaire O, Llurba E, Ramoni A, Vatish M, Wertaschnigg D, Galindo A: Implementation of the sFlt-1/PlGF ratio for prediction and diagnosis of preeclampsia in singleton pregnancy: implications for clinical practice. Ultrasound ObstetGynecol 2015;45:241–246 [55]. (2) Herraiz I, Llurba E, Verlohren S, Galindo A. Spanish Group for the study of angiogenic markers in preeclampsia. Update on the diagnosis and prognosis of preeclampsia with the aid of the sFlt-1/PlGF ratio in singleton pregnancies. FetalDiagnTher 2018;43:81–89 [56].

**Table 8 ijms-24-00972-t008:** Distributions of the values of sFlt-1, sEng, PlGF and the sFlt-1/PlGF and sFlt-1*sEng/PlGF ratios in pregnancies with and without adverse outcomes.

		Mean	Median	Q1	Q3	SD
sEng [ng/mL]	adverse	11.6	11.8	11.3	12.2	1
	non adverse	10.4	11.8	9.8	12.1	2.8
PIGF [pg/mL]	adverse	107	58	25	186	100
	non adverse	156	123	26	190	189
sFlt-1 [pg/mL]	adverse	108,727	77,932	6201	156,292	124,261
	non adverse	82,512	48,260	13,052	135,395	87,355
Ratio sFlt-1/PlGF	adverse	2610	476	130	1032	5544
	non adverse	800	322	143	1090	982
Ratio sFlt*sEng/PlGF	adverse	31,599	5876	1560	12,714	66,852
	non adverse	9527	3423	1493	12,996	12,084
Gestational age [weeks]	adverse	31	32	27	35	4.8
	non adverse	35	35	33	37	2.7

**Table 9 ijms-24-00972-t009:** Comparison of the results in early and late PE.

	Early PE <34 Weeks		Late PE ≥34 Weeks		*p* Value
	*n* = 22		*n* = 21		
	Median	Q1–Q3	Median	Q1–Q3	
Gestational age [weeks]	31.5	27–32	36	35–37	0.0000001
PlGF pg/mL [pg/mL]	33	21–125	124	36–238	0.01
sFlt-1 [pg/mL]	104,310	3152–172,024	74,759	14,981–135,395	NS
sEng [ng/mL]	12	11.6–12.2	11.7	10.2–12.1	NS
sFlt-1/PlGF	973	228–2247	364	228–960	NS

NS stands for Not Statistically Significant.

**Table 10 ijms-24-00972-t010:** Comparison of the results in early and late FGR.

	Early FGR <34 Weeks		Late FGR ≥34 Weeks		*p* Value
	*n* = 5		*n* = 9		
	Median	Q1–Q3	Median	Q1–Q3	
Gestational age [weeks]	31	27–32	36	35–37	0.0000001
PlGF [pg/mL]	119	117–150	172	143–221	0.01
sFlt-1 [pg/mL]	115,490	56,238–127,053	29,183	13,051–36,878	<0.05
sEng [ng/mL]	11.7	11.6–12	10.9	9.2–11.8	NS
sFlt-1/PlGF	972	845–984	212	80–222	0.01

NS stands for Not Statistically Significant.

**Table 11 ijms-24-00972-t011:** Preeclampsia diagnostic criteria formed by International Society for the Study of Hypertension in Pregnancy (ISSHP) in 2018.

Preeclampsia
Preeclampsia is gestational hypertension accompanied by ≥1 of the following new-onset conditions at or after 20 weeks’ gestation:
Proteinuria
Other maternal organ dysfunction, including:
AKI (creatinine ≥ 90 umol/L; 1 mg/dL)
Liver involvement (elevated transaminases, e.g., alanine aminotransferase or aspartate aminotransferase >40 IU/L) with or without right upper quadrant or epigastric abdominal pain
Neurological complications (examples include eclampsia, altered mental status, blindness, stroke, clonus, severe headaches, and persistent visual scotomata)
Hematological complications (thrombocytopenia–platelet count <150,000/μL, disseminated intravascular coagulation, hemolysis)
Uteroplacental dysfunction (such as fetal growth restriction, abnormal umbilical artery [UA] Doppler wave form analysis, or stillbirth)

**Table 12 ijms-24-00972-t012:** Consensus-based definitions for early and late fetal growth restriction (FGR).

Early FGR: GA <32 weeks, in absence of congenital anomalies	Late FGR: GA ≥32 weeks, in absence of congenital anomalies
AC/EFW < 3rd centile or UA-AEDF Or 1. AC/EFW < 10th centile combined with 2. UtA-PI > 95th centile and/or 3. UA-PI >95th centile	AC/EFW < 3rd centile Or at least two out of three of the following 1. AC/EFW < 10th centile 2. AC/EFW crossing centiles >2 quartiles on growth centiles * 3. CPR < 5th centile or UA-PI > 95th centile

AC, fetal abdominal circumference; AEDF, absent end-diastolic flow; CPR, cerebroplacental ratio; EFW, estimated fetal weight; GA, gestational age; PI, pulsatility index; UA, umbilical artery; UtA, uterine artery. Adapted from: Gordijn SJ, Beune IM, Thilaganathan B, Papageorghiou A, Baschat AA, Baker PN, Silver RM, Wynia K, Ganzevoort W. Consensus definition of fetal growth restriction: a Delphi procedure. Ultrasound Obstet Gynecol. 2016;48:333–9. AC, fetal abdominal circumference; AEDF, absent end-diastolic flow; CPR, cerebroplacental ratio; EFW, estimated fetal weight; GA, gestational age; PI, pulsatility index; UA, umbilical artery; UtA, uterine artery [83]. * Hadlock FP, et al., In utero analysis of fetal growth: a sonographic weight standard. Radiology. 1991 Oct;181(1):129–33 [53].

**Table 13 ijms-24-00972-t013:** Summary of the recommendations for the use of sFlt-1/PlGF ratio in women with signs and symptoms of PE based on the opinion of experts in the use of angiogenic markers.

Ratio sFlt-1/PlGF (EP/LP)	Interpretation	Time to Delivery (ep)	What Should Be Done?
LOW: <38	Rule out PE: 1 week: NPV ≈ 99% 4 weeks: NPV ≈ 95%	Unmodified	Reassuring the patient. No further determinations are needed unless new suspicion arises.
Intermediate: 38–85/38–110	Rule in PE: 4 weeks PPV ≈ 40%	20% remain pregnant after 1 month	Follow up visit and retest in 1–2 weeks. Maternal education about signs and symptoms of PE
High: >85/>110	Diagnosis of PE (or PD-related disorder) is highly likely	15% remain pregnant after 2 weeks	Follow up visit and retest in 2–4 days. EP: consider referral to higher-level center. LP: consider lowering the threshold for labour induction.
Very high: >655/>201	Short-term complications and need to deliver are highly likely	30% remain pregnant after 2 days	Close surveillance. EP: corticosteroids to the mother for fetal lung maturation

NPV, ang. negative predictive value; PD, placental dysfunction; PE, preeclampsia; PlGF, placental growth factor; PPV, positive predictive value; sFlt-1, soluble form of fms-like tyrosine kinase-1; EP, early phase (<34 weeks); LP, late phase (≥34 weeks).

## Data Availability

Data available on request due to restrictions e.g., privacy or ethical. The data presented in this study are available on request from the corresponding author.

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
