# Peer review of "Similar Pro- and Antiangiogenic Profiles Close to Delivery in Different Clinical Presentations of Two Pregnancy Syndromes: Preeclampsia and Fetal Growth Restriction"

_ijms, 2023, doi:10.3390/ijms24020972_

Round 1

Reviewer 1 Report

The manuscript by Dymara-Konopka et al., aims to evaluate serum levels of anti- and pro-angiogenic proteins in pregnancies complicated preeclampsia and/or fetal growth restriction.

I acknowledge to the authors that they have clearly spent a lot of time compiling this manuscript, there are 150 references and 11 tables presented. However, I have not been convinced of the relevance of analyzing PlGF and sFLT at time of delivery, or how the information presented is novel.

Also, I am confused as to how the ‘healthy control pregnancies’ were obtained.

Line 769 the authors state “The entire study group was pairwise matched by gestational age with healthy control pregnancies (n = 20).”

Line 826 “control group included only women with delivery of a term (>37 weeks) infant whose birth weight was between the 10th and 90th percentile for gestational age and no medical, obstetrical nor surgical complications during the entire gestation.”

However, the gestation of the preeclampsia and FGR groups range from 32-35 weeks so how can the controls be >37 weeks but also pairwise matched by gestational age?

Also, in Table 1, the control group has a gestation (weeks) of Q1-Q3: 31-37, so there were some preterm deliveries?

And Table 4, presents control group median as 34, yet the Q1-Q3 is 37-39, so cannot have a median of 34.

Author Response

Reply to Reviewer 1

Thank you very much for bringing this to my attention and for this comment.

According to your suggestions, some changes in the paper were introduced.

1/ Reviewer : “The manuscript by Dymara-Konopka et al., aims to evaluate serum levels of anti- and pro-angiogenic proteins in pregnancies complicated preeclampsia and/or fetal growth restriction.

I acknowledge to the authors that they have clearly spent a lot of time compiling this manuscript, there are 150 references and 11 tables presented. However, I have not been convinced of the relevance of analyzing PlGF and sFLT at time of delivery, or how the information presented is novel.

Also, I am confused as to how the ‘healthy control pregnancies’ were obtained.

Line 769 the authors state “The entire study group was pairwise matched by gestational age with healthy control pregnancies (n = 20).”

Line 826 “control group included only women with delivery of a term (>37 weeks) infant whose birth weight was between the 10th and 90th percentile for gestational age and no medical, obstetrical nor surgical complications during the entire gestation.”

However, the gestation of the preeclampsia and FGR groups range from 32-35 weeks so how can the controls be >37 weeks but also pairwise matched by gestational age?

Also, in Table 1, the control group has a gestation (weeks) of Q1-Q3: 31-37, so there were some preterm deliveries?

And Table 4, presents control group median as 34, yet the Q1-Q3 is 37-39, so cannot have a median of 34.

Dear Reviewer,

  • Thank you very much for bringing the topic of matching the control group to my attention and for the entire comment. Regarding the considerations about gestational age in the control group, we kindly explain:
  1. There were a prospective case-control study was performed consisting of 77 singleton pregnancies complicated by preeclampsia and / or FGR pairwise matched by gestational age with healthy pregnancies at the moment of blood sampling (no at the time of labour in the control group. It was max. 48 hrs before indicated delivery [necessary birth] in all patients with pregnancy complications such as FGR or severe preeclampsia when was indication for delivery due to deterioration of the mother's condition or fetal risk. But not in the control subjects. All studied patients from the control group delivered at term.

  1. Healthy, non-complicated pregnant patients from our outpatient department have had the venipuncture performed at the gestational age which corresponded with gestational age of patients from the study group (iFGR, iPE, PE+FGR).

  1. There were initially more patients assigned to the control group, who had their blood taken, that were finally included in the control group. Patients, who at the end of their pregnancy presented any complications (like gestational diabetes), or delivered preterm / SGA baby were excluded from the study. Retrospective analysis of perinatal results in patients initially assigned to the control group was performed. Patients who did not fulfill the criteria were excluded from the study.

  1. Q1-Q3 31-37 in Table 1 refers to the gestational age of blood sampling.

  • In FGR /PE groups this occurred within 48hrs before the delivery.
  • In control groups patients, gestational age at sampling was pairwise matched with patients from investigated PE/FGR groups together.
  • Thus, there were no preterm deliveries in the control group. „Gestational age at birth (weeks)” presented in Table 7 refers to the gestational age of delivery, which in case of control group patients was always at term (37+0 or more).
  • The delivery of patients in the control group occurred weeks after blood sampling that took place at earlier gestational age.

  • We tried to do our best to improve the style and spelling of English language.

The manuscript was reviewed and corrected by an English native speaker

Thank You again for Your help.

Yours faithfully,

Reviewer 2 Report

Dear Authors,

Currently, the number of studies on the prediction of adverse outcomes associated with placental dysfunction based on the measurement of the pro- and anti-angiogenic markers is increasing. The authors also undertook studies of the pro- and antiangiogenic profile in the serum of pregnant women with preeclampsia and fetal growth restriction. It should be noted that the authors clearly formed clinical groups in accordance with accepted diagnostic criteria. However, there are a few limitations, which were also pointed out by the authors.

One of them concerns the lack of division into late-onset and early-onset of the above syndromes. Early-onset and late-onset severe preeclampsia are known to have different molecular mechanisms, whereas the late-onset mild preeclampsia may have no placenta-specific causal factors. The origin of late-onset preeclampsia may be dominated by predisposing cardiovascular or metabolic risks of endothelial dysfunction as part of a systemic inflammatory response. In this regard, is it correct to consider the results as diagnostic criteria for both preeclampsia and FGR without division into terms?

In my opinion, the authors also need to highlight / emphasize the advantages of their study, since it is not entirely clear from the Discussion what the authors offer that is new in comparison with similar studies conducted by other researchers.

There are a few comments below.

Table 3 does not indicate the p-values. It is necessary to indicate the decoding of the abbreviations of the ultrasound data at the bottom of the table

There is no reference in the text to table 8. I don’t quite understand if there is a need for it at all, if all the distribution information is already indicated in table 7?

Discussion.

Please indicate references to articles at the beginning of the Discussion, where these statements are postulated.

This section is duplicated.  “It was postulated that maternal PE arises from soluble factors released into sys- 285 temic circulation from the stressed placenta. Fetoplacental hypoxia is one of the conse- 286 quences of placental impairment, which is a threat to the survival of the fetus. It was pos- 287 tulated that maternal PE arises from soluble factors released into systemic circulation from 288 the stressed placenta. Fetoplacental hypoxia is one of the consequences of placental im- 289 pairment, which is a threat to the survival of the fetus”.

In general, the Discussion is too long and written in the style of a review. It is unacceptable. It is necessary to make it short and indicate only the key theses that were obtained as a result of your experiment and what is their originality in comparison with the studies of other authors.

Author Response

Reply to Reviewer 2

Dear Reviewer,

Thank you very much for bringing this to my attention and for this comment.

According to your suggestions, some changes in the paper were introduced.

Few specific comments

1/ Reviewer:Currently, the number of studies on the prediction of adverse outcomes associated with placental dysfunction based on the measurement of the pro- and anti-angiogenic markers is increasing. The authors also undertook studies of the pro- and antiangiogenic profile in the serum of pregnant women with preeclampsia and fetal growth restriction. It should be noted that the authors clearly formed clinical groups in accordance with accepted diagnostic criteria. However, there are a few limitations, which were also pointed out by the authors.

One of them concerns the lack of division into late-onset and early-onset of the above syndromes.”

Answer :

  • Thank you for bringing this to our attention. We have added the table 9 and table 10 to the manuscript to address this concern.
  • The results of our studies - obtained results of angiogenic factors - in groups divided into early and late preeclampsia and early and late FGR were added to the manuscript as suggested by the Reviewer and described in table 9 and table 10.

2/ Reviewer: “In my opinion, the authors also need to highlight / emphasize the advantages of their study.”

Thank you for bringing this to our attention.

  • The entire study cohort was analyzed with respect to adverse pregnancy outcomes that occurred. In all investigated subgroups PlGF was lower and sEng, the sFlt-1/PlGF and sFlt-1*sEng/PlGF ratio were higher than in the control group. Differences were most strongly pronounced in the PE with concurrent FGR group and in the sFlt-1/PlGF ratio. The highest sFlt-1 values in preeclamptic patients suggest that this substance may be responsible for reaching the threshold needed for PE to develop as a maternal manifestation of ischemic placental disease. FGR is characterized by elevated maternal sFlt-1/PlGF ratio, which boosts at the moment of indicated delivery due to fetal risk.
  • We concluded that angiogenic imbalance is reflective of placental disease regardless of its clinical manifestation in the mother, and may be used as support for the diagnosis and prognosis of FGR.
  • We also observed that Forty-eight hours before delivery, elevation of the sFlt-1 to PlGF ratio in iFGR, iPE and PE+FGR patients reached extreme values exceeding the cutoff of 655/201 in most women (64%, 65% and 70% respectively) with similar occurrence in all investigated subgroups. The results are presented in Table 6.
  •  

3/ Reviewer: Table 3 does not indicate the p-values. It is necessary to indicate the decoding of the abbreviations of the ultrasound data at the bottom of the table.”

Thank you for bringing this to our attention.

Table 3 was modified: p values were added and abbreviations of ultrasound parameters attached. 

4/ Reviewer : “There is no reference in the text to table 8. I don’t quite understand if there is a need for it at all, if all the distribution information is already indicated in table 7?”

Thank you for bringing this to our attention.

  • Table 8 is in our opinion needed as it presents not only the sFlt-1/PlGF ratio values, but also concentrations of all the measured substances separately: sEng, sFlt-1, PlGF in the adverse and no-adverse group.
  • Appropriate reference was added in the text: Chapter 2.6 Adverse pregnancy outcomes: In the end of the last sentence: At the moment of prompt delivery, there were no statistical differences in terms of angiogenic factors levels nor their ratios between the adverse and no-adverse groups (Table 8).

5/ Reviewer: “Discussion.

  • Please indicate references to articles at the beginning of the Discussion, where these statements are postulated.
  • This section is duplicated. “It was postulated that maternal PE arises from soluble factors released into systemic circulation from the stressed placenta. Fetoplacental hypoxia is one of the consequences of placental impairment, which is a threat to the survival of the fetus. It was postulated that maternal PE arises from soluble factors released into systemic circulation from the stressed placenta. Fetoplacental hypoxia is one of the consequences of placental impairment, which is a threat to the survival of the fetus”.
  • In general, the Discussion is too long and written in the style of a review.”

Thank you very much for this comment.

  • We have added references to these sentences at the beginning of the Discussion, where this statements are postulated.
  • The duplicated sentences have been removed from the manuscript.
  • Discussion was edited and also shortened as suggested by the Reviewer.
  • We have modified the entire manuscript to address this concern.

Thank You again for Your help.

Your faithfully,

Marzena Laskowska

Authors

Round 2

Reviewer 2 Report

Dear Authors,

In the previous version of the manuscript, the tables had a more compact size. The drawings were also located side by side. Bring it into line. Besides, why are the table names below the tables? It needs to be corrected.

It's great that Authors added tables 9 and 10, however, I did not find a link to them in the text and description.

Concerning to the Discussion. We must give credit to authors have significantly revised it. However, as I noted earlier, the style of the Discussion is very similar to a Review. And the results of the authors are simply scattered throughout in the text. The authors provide so many descriptions of studies done by other authors. Definitely, it is necessary to describe your own results in comparison with the results of other research. But, in my opinion, the authors need to highlight the key points! of their own research and, in brief, in comparison with similar studies of other authors, provide a description of them, but without detailing the studies of others. Otherwise, the essence of your work is fragmented and incomprehensible. In the Discussion, it is enough to leave 3-4 sheets in order to focus on your results in the context of data from other authors.

However, as I noted earlier, the style of the discussion is very similar to the review. And the results of the authors are simply scattered throughout the text. The authors provide so many descriptions of studies done by other authors. Definitely, it is necessary to describe your own results in comparison with the results of other authors. But, in my opinion, the authors need to highlight the key points! own research and, in brief, in comparison with similar studies of other authors, give a description of them, but without detailing the studies of others. Otherwise, the essence of your work is fragmented and incomprehensible. In the Discussion, it is enough to leave 3-4 sheets in order to emphasize on your results in the context of data from other authors.

Author Response

Thank you very much for bringing this to my attention and for this comment.

According to your suggestions, some changes in the paper were introduced.

Few specific comments

1/ Reviewer:In the previous version of the manuscript, the tables had a more compact size. The drawings were also located side by side. Bring it into line. Besides, why are the table names below the tables? It needs to be corrected.

Answer :

  • Thank you for bringing this to our attention.

 It was improved and corrected to the Reviewer's suggestions.

2/ Reviewer:

It's great that Authors added tables 9 and 10, however, I did not find a link to them in the text and description.

Answer :

The links to tables 9 and 10 has been added.

Thank you very much for bringing this to my attention and for this comment.

3/ Reviewer: Concerning to the Discussion. We must give credit to authors have significantly revised it. However, as I noted earlier, the style of the Discussion is very similar to a Review. And the results of the authors are simply scattered throughout in the text. The authors provide so many descriptions of studies done by other authors.   In the Discussion, it is enough to leave 3-4 sheets in order to emphasize on your results in the context of data from other authors.

Answer :

The Discussion section has been improved and shortened according to the Reviewer's suggestions. Thank you for drawing our attention.

Thank You again for Your help.

Your faithfully,

Marzena Laskowska

Authors

Round 3

Reviewer 2 Report

The authors have done an excellent job of editing the Discussion.

Author Response

Dear Editor

Thank you very much for bringing this to my attention and for this comment.

 The title has been adapted to:

"Similar pro- and antiangiogenic profiles close to delivery in different

clinical presentations of two pregnancy syndromes: preeclampsia and fetal

growth restriction".

According to the Editor’s suggestion

Thank You again for Your help.

Your faithfully,

Marzena Laskowska

Authors
